# Selectively Modified Lactose and *N*-Acetyllactosamine Analogs at Three Key Positions to Afford Effective Galectin-3 Ligands [note 1]

**DOI:** 10.3390/ijms24043718

**Published:** 2023-02-13

**Authors:** Shuay Abdullayev, Priyanka Kadav, Purnima Bandyopadhyay, Francisco Javier Medrano, Gabriel A. Rabinovich, Tarun K. Dam, Antonio Romero, René Roy

**Affiliations:** 1Glycosciences and Nanomaterials Laboratory, Université du Québec à Montréal, Succ. Centre-Ville, P.O. Box 8888, Montréal, QC H3C 3P8, Canada; 2Laboratory of Mechanistic Glycobiology, Department of Chemistry, Michigan Technological University, 1400 Townsend Drive, Houghton, MI 49931, USA; 3Centro de Investigaciones Biológicas “Margarita Salas” (CIB), CSIC, E-28040 Madrid, Spain; 4Laboratorio de Glicomedicina, Instituto de Biología y Medicina Experimental (IBYME), Consejo Nacional de Investigaciones Científicas y Técnicas (CONICET), Vuelta de Obligado 2490, C1428 Ciudad de Buenos Aires, Argentina

**Keywords:** galectin, lactoside, isothermal titration calorimetry (ITC), X-ray

## Abstract

Galectins constitute a family of galactose-binding lectins overly expressed in the tumor microenvironment as well as in innate and adaptive immune cells, in inflammatory diseases. Lactose ((β-D-galactopyranosyl)-(1→4)-β-D-glucopyranose, Lac) and *N*-Acetyllactosamine (2-acetamido-2-deoxy-4-*O*-β-D-galactopyranosyl-D-glucopyranose, LacNAc) have been widely exploited as ligands for a wide range of galectins, sometimes with modest selectivity. Even though several chemical modifications at single positions of the sugar rings have been applied to these ligands, very few examples combined the simultaneous modifications at key positions known to increase both affinity and selectivity. We report herein combined modifications at the anomeric position, C-2, and *O*-3′ of each of the two sugars, resulting in a 3′-*O*-sulfated LacNAc analog having a Kd of 14.7 µM against human Gal-3 as measured by isothermal titration calorimetry (ITC). This represents a six-fold increase in affinity when compared to methyl β-D-lactoside having a Kd of 91 µM. The three best compounds contained sulfate groups at the *O*-3′ position of the galactoside moieties, which were perfectly in line with the observed highly cationic character of the human Gal-3 binding site shown by the co-crystal of one of the best candidates of the LacNAc series.

## 1. Introduction

Galectins are evolutionarily conserved, soluble glycan-binding proteins (lectins) found in a wide variety of taxonomic groups. Galectin-3 (Gal-3) is an exceptional member of the galectin family because of its unique structure and functional diversity. Unlike other galectins, Gal-3 is a chimeric protein composed of distinct domains. In human Gal-3, the C-terminal carbohydrate recognition domain (CRD) is connected to an N-terminal tail (NT). The NT possesses nine non-triple-helical collagen-like Pro/Gly-rich repeats followed by a 21-amino-acid-long N-terminal segment [1]. While the CRD contains the glycan-binding site, the non-lectin NT domain contributes to the oligomerization of this lectin [2].

Galectin-3 contributes to various pathophysiological processes, including inflammation and fibrosis, cancer cell proliferation, adhesion, angiogenesis, cell migration, T-lymphocytes apoptosis and macrophage differentiation into infiltrative forms that stabilize tumor environments and polarization of tumor-associated macrophages [3,4,5,6,7,8]. Because of its disease-specific overexpression, human Gal-3 (hGal-3) has been proposed to be a therapeutic target and a biomarker of different pathological conditions including cancer and cardiovascular diseases [9,10].

The glycan-binding site of Gal-3, located on the CRD, recognizes galactosyl moieties, primarily lactose ((β-D-galactopyranosyl)-(1→4)-β-D-glucopyranose, Lac) residues. Such residues abundantly occur on glycoproteins associated with cell surfaces and the extracellular matrix. Most reported biological functions of Gal-3 have been shown to depend on its binding to N- and O-linked glycans of glycoproteins. The location and orientation of three specific hydroxyl groups (4-OH, 6-OH of Gal, and 3-OH of glucose/N-acetylglucosamine) of lactose/LacNAc were reported to be crucial for creating bonds with the amino acids in the vicinity of the glycan-binding site of Gal-3 [11]. However, this requirement appears to be flexible, as Gal-3 shows remarkable plasticity in ligand recognition. For example, the binding site of Gal-3 accommodates a sialic acid residue (as in α2,3-sialyllactose) via its so-called extended binding pocket [12]. The lectin also binds to galactomannans and polymannan [13]. The diversity of Gal-3 ligands was further expanded with the finding that Gal-3 could interact with sulfated glycosaminoglycans (GAGs) and chondroitin sulfate proteoglycans (CSPGs) through its lactose/LacNAc-binding site [11]. The sulfated GAG-binding ability of Gal-3 indicates that its binding site can tolerate non-lactosyl moieties as well as sulfate groups.

Consequently, the design of highly selective sugar-based inhibitors against each individual member of the galectin family has been the subject of intense research activities [14,15,16,17,18,19,20,21,22,23]. Essentially based on β-D-galactopyranoside-based leads, the incorporation of aglyconic pharmacophores has provided successful candidates. Moreover, modifications at position -3 of the galactopyranosides residue or *O*-3′ in the case of lactosides provided additional key binding interactions. Appealingly, the introduction of a negative sulfate group at *O*-3/*O*-3′ has also afforded potent ligands owing to the presence of charged amino acids within the CRD. Indeed, a successful glycomimetic has reached clinical phases [18]. In addition to the above chemical modifications onto monomeric antagonists, multivalent galactosides/lactosides have similarly provided substantial gains in both affinity and selectivity [24,25,26]. This was predominantly important for the chimeric Gal-3 that can oligomerize upon binding to multivalent receptors due to its collagenous peptide tail [27]. In the present study, we synthesized a family of monomeric lactoside/N-acetyllactosamine analogs modified at three key positions known to alter the putative binding interactions with Gal-3.

## 2. Results and Discussion

### 2.1. Synthesis

This work was undertaken to further explore the binding interactions between human galectin-3 (hGal-3) and β-D-galactopyranosides containing pharmacophores based on X-Ray crystallographic data [28].

To this end, three families of derivatives were synthesized. In the first two cases, lactosides (Lac) (Figure 1) and *N*-acetyllactosamines (LacNAc) (Figure 2, Figure 3 and Figure 4), modified at three key positions, were prepared, while in the third case, a sialyl-α-(2-3)-LacNAc trisaccharide (Figure 4) was considered because of the extended binding site capable of accommodating sialylated oligosaccharides [12].

Figure 1 illustrates the syntheses of propargyl- (**4**) and *para*-nitrophenyl β-D-lactopyranosides (**6**), together with their 3′-*O*-sulfated analogs **11** and **12**, respectively. Methyl β-D-lactopyranoside (**1**) was used as a reference compound. Hence, per-*O*-acetylated lactose (**2**) was propargylated under previously published conditions (propargyl alcohol, BF_3_-Et_2_O, DCM) to provide lactoside**3**, which was uneventfully de-*O*-acetylated (NaOMe, MeOH) to give propargyl β-D-lactopyranoside (**4**) (Figure 1) [29]. Alternatively, the corresponding *para*-nitrophenyl (PNP) analog (**5**) and its de-*O*-acetylated version (**6**) were prepared according to our previously published phase-transfer catalyzed (PTC) procedure [15]. The direct incorporation of a 3′-*O*-sulfate group to lactoside through tin acetal chemistry is a well-documented approach. However, previous investigations were shown to afford mixtures of 3′- as well as 6′-*O*-sulfates [23,30] that, in our hands, proved difficult to separate. The problem was readily solved by first preparing 6-, 6′-bulky silyl protecting groups (TBDPSCl, pyridine, r.t., 8 h.) which gave lactoside intermediates **7** and **8** in high yields. Cyclic tin acetylation (Bu_2_SnO, MeOH, 80 °C, 4 h.), followed by sulfation using the Et_3_N-SO_3_ complex (DMF, 60 °C, 17 h), afforded compound **9** and **10** in 88 and 86% yields, respectively. Removal of silyl protecting groups (HF-pyridine, 0 °C to r.t., 24 h), gave the desired 3′-*O*-sulfated lactosides **11** and **12** in good yields.

Even though there are several syntheses of β-D-LacNAc glycosides, most following complex-protecting group manipulations, a simplified approach was followed that was based on previous observations, leading to the regioselective galactosylation of a 6-*O*-TBDPS-protected *N*-acetylglucosamine glycoside. Thus, starting with the known *N*-acetyl-2-amino-2-deoxy-3,4,6-tri-*O*-acetyl-α-D-glucopyranosyl chloride **13** (Figure 2), the corresponding *para*-nitrophenyl 2-acetamido-3,4,6-tri-*O*-acetyl-2-deoxy-β-D-glucopyranoside **14** was prepared under the above PTC conditions [31]. Following usual transesterification under Zemplén conditions (NaOMe, MeOH) gave **15** quantitatively. 6-*O*-Silylation as above (TBDPSCl, pyridine, r.t., 8 h) provided **16** in 91% yield. Regioselective galactosylation with trichloacetimidate donor **17** (BF_3_-Et_2_O, DCM/THF, -35°C, 5 h.) afforded 4-*O*-galactosylated PNP LacNAc congener **18** in 38% yield, together with recovered glycosyl acceptor **16** (~30%). The low yield obtained, in comparison to previously published case [19], was attributed to the lower solubility of **16** under the reaction conditions.

The structure of **18** was fully characterized using ^1^H- and ^13^C-NMR techniques (see Appendix A). The stereo- and regioselectivity of the formed interglycosidic linkage was unambiguously determined by a HMBC experiment, where a *H*1 (Gal)–*C*4 (Glc) correlation could be readily observed (Appendix A). Deacetylation of **18** as above (0.5 eq NaOMe in MeOH/THF mixture at room temperature, 3.5 h of stirring) afforded **19** in 97% yield. Silyl group deprotection (HF-pyridine, 0 °C to r.t., 24 h), provided PNP-LacNAc**20** in 91% yield. Repeating the above 3′-*O*-sulfation strategy, i.e., protection of *O*-6′ with TBDPSi ether (**21**, 94%), dibutylstannylene acetal-mediated regioselective sulfation at *O*-3′, followed by desilylation gave the desired *para*-nitrophenyl-based 3′-*O*-sulfo-LacNAc intermediate **22** which was achieved in 53% yield over three steps. Both silyl groups removal as above (HF-pyridine, 0 °C to r.t., 24 h) furnished **23** in 72% yield.

A similar synthetic strategy en route toward the analogous propargylated LacNAc derivative **27** was followed, as described in Figure 3. Therefore, *N*-acetylglucosamine **24** was transformed into protected intermediate **25 [19]**, which upon full deprotection (TBAF, THF, r.t., 14 h; then NaOMe, MeOH, r.t., 3 h) gave **27** in 88% yield. Otherwise, Zemplén transesterification of **25** as above gave **26** quantitatively that was silylated at *O*-6′ (TBDPSCl, pyridine, r.t., 8 h) to give **28** in 96% yield.

The syntheses of the remaining propargylated LacNAc-based derivatives, having anionic groups at the key 3′-position, are illustrated in Figure 4. Hence, bis-6,6′-*O*-silyl LacNAc intermediate **28** was first sulfated at the 3′ position following the above tin acetal procedure (Bu_2_SnO, MeOH, 80 °C, 4 h) followed by sulfation using the Et_3_N-SO_3_ complex (DMF, 60 °C, 17 h, 90% overall), which gave **29**, which after silyl group deprotection as before, provided target compound **30** in 64% yield. To fully exploit the highly cationic character of the hGal-3 deep binding pocket, a carboxylate derivative was also prepared. In this case, compound **28** was treated under the tin acetal conditions described above but using *tert*-butyl bromoacetate as electrophile which afforded ester **31** in 80% yield. Acid deprotection of the *t*-butyl ester (TFA, DCM, r.t., 0.5 h) followed by desilylation and neutralization gave **32** (85%). The synthesis of the sialylated trisaccharide **33** (Appendix A) has been recently described [32].

**Scheme 4 ijms-24-03718-sch004:**
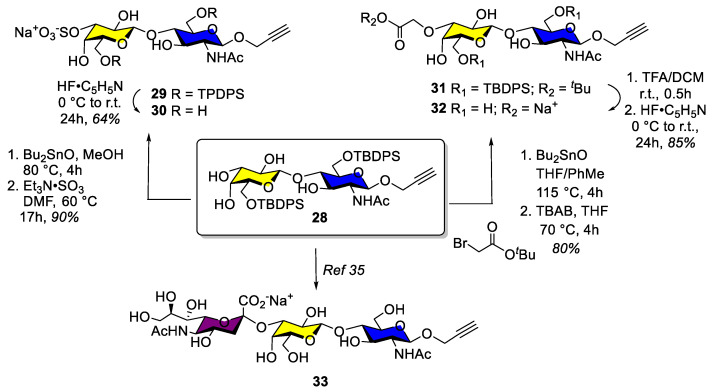
Tin acetal catalyzed regioselective 3′-*O*-sulfation (**30**), 3′-*O*- etherification (**32**), and 3′-*O* sialylation following a recently described strategy (**33**) of intermediate 28 [32].

### 2.2. Isothermal Titration Calorimetry

Binding affinities (Kd values) and stoichiometry (n value) of hGal-3 for methyl lactoside (**1**) and some strategically designed analogs of lactose (**4**, **6**, **11** and **12**) as well as LacNAc (**20**, **23**, **27**, **30**, **32**, **33**) were measured by isothermal titration calorimetry (ITC). All the analogs showed “n” values close to 1.0, indicating that they bound to a single binding site on the hGal-3 monomer (Figure 1). When interacted with Gal-3, all the analogs tested in this study showed higher affinities compared to methyl lactoside (**1**). The Kd value of **1** for Gal-3 was 91.0 µM. Among all the analogs, 3′-*O*-sulfated propargylated LacNAc (**30)** showed the highest affinity (Kd = 14.7 µM) for hGal-3. 3′- *O*-sulfated PNP lactose (**12**) and 3′-O-sulfated PNP LacNAc (**23**) possessed the second (Kd = 20.4 µM) and third highest (Kd = 22.2 µM) affinity, respectively. The Kd value of 3′-*O*-sulfated propargylated lactose (**11**) was 37.0 µM, which indicates that the absence of the *N*-acetyl group (when compared with **30**) or the replacement of the PNP group (when compared with **12** and **23**) in this propargylated analog (**11**) slightly reduced its affinity for hGal-3. The Kd value of PNP LacNAc (**20**) was similar to that of its propargyl Lac analog **11**.

Removal of 3′- *O*-sulfate group from lactoside analogs (as in **4** and **6**) and LacNAc analogs (as in **27** and **32**) reduced their affinity for hGal-3. The Kd values of **4** and **6** were 47.6 µM and 50.0 µM, respectively. Analogs **27** and **32** showed Kd values of 38.0 µM and 52.6 µM, respectively. These results suggest that the binding site of hGal-3 can beneficially accommodate an anionic sulfate group. This observation is consistent with a report that hGal-3 recognizes sulfated glycosaminoglycans (GAGs) [11]. Moreover, the anionic characters of the 3′-*O*-sulfated analogs in both series can clearly compensate for the cationic environment of the hGal-3 binding site near the 3′-position (see X-Ray analysis below).

Unlike galectin-1 (Gal-1), hGal-3 can tolerate terminal sialic acid of a glycan chain. In the present study, when the propargylated LacNAc (**27**) contained a terminal sialic acid residue (as in analog **33**), the affinity was slightly reduced (compared to **27**), but hGal-3 could still recognize the sialylated analog **33.** The -TΔS values (Table 1) suggest that entropy played small but favorable roles in the binding of all the analogs, except **20**, **32** and **33**, where enthalpies were relatively dominant and the entropies were unfavorable. Entropic contributions were comparatively higher in the binding of **6**, **11**, and **23**. This observation is in line with previously published data concerning the important role of conformational entropy and solvation entropy in protein–ligand binding [22] (see difference between **4** and **6**). Overall, the ITC data showed that the binding site hGal-3 can accommodate different variations of the parent lactosyl moiety and that strategic modifications of lactose and LacNAc can increase their binding affinity for hGal-3.

### 2.3. Structure of Gal-3C in Complex with ***23***

This is the first structure to be reported of a sulfated sugar bound to the active site of the Gal-3 CRD. The molecular structure of the sugar, **23**, used in this study is shown in Figure 2.

Compound **23** is bound to the active site of the CRD in a very similar manner as lactose and LacNAc (Figure 2a). Residues His158, Asn160, Arg162, Asn174, Trp181 and Glu184 are involved in direct contacts, and residues Arg144, Asp148 and Glu165 make contact through waters. Figure 2a shows the electron density map for the sugar bound to the Gal-3 CRD active site. There is very good electron density for the sulfate and the two sugar moieties, while the nitro-phenyl substituent at *O*1′ does not show nearly any electron density. This is reflected on the B factor distribution of the molecule **23** (Figure 2b) that shows the galactopyranose moiety firmly anchored to the active site with an average B factor of 17.42 Å^2^. The sulfate group shows a higher average B factor of 32.77 Å^2^, reflecting no direct contact with the protein. Meanwhile, the N-acetyl glucopyranose shows a more relaxed binding with an average B factor of 52.58 Å^2^. Finally, the nitro-phenyl substituent does not have any contact with protein residues and shows a very high B factor average of 178.97 Å^2^.

The typical interactions of the galactopyranose and the glucopyranose moieties of **23** are very similar to those of lactose. The sulfate group contacts the protein through water molecules; one oxygen interacts with one water (S1) that is in contact with Trp181, and a second oxygen contacts two waters (S2 and S3) that are in contact with Asp148 (Figure 3a). Also, the oxygen of the *N*-acetyl substituent in the glucopyranose moiety makes one direct contact with Glu184.

The superposition of all Gal-3 CRD structures present in the PDB in complex with lactose, LacNAc (β-D-galactopyranose-(1-4)-2-acetamido-2-deoxy-β-D-glucopyranose) (PDB ID 1KJL) [33] and LacNAcNAc (β-D-galactopyranose-(1-4)-*N*-acetyl-2-(acetyla-mino)-2-deoxy-β-D-glucopyranosylamine) (PDB ID 5NF7) [34,35], is shown in Figure 3b. Most of the residues in the binding site are in very similar positions; only Arg144 shows different conformations. Moreover, the conserved water network adding to the binding of sugar can be observed in Figure 3b. Seven waters are conserved in our structure (S1 through S7), water 8 (S8) is missing in our structure, and one water (S9) has been replaced by one of the oxygens from the sulfate group.

The superposition of all Gal-3 CRD structures with a sugar containing a *N*-acetyl substituent at *O*2′ are shown in Figure 3c. There are only three structures in the PDB, two in complex with LacNAc and one in complex with LacNAcNAc. The position of this group is not very well fixed. In two of the structures the oxygen does not appear to contact Glu184. One structure makes a water (S10)-mediated contact with this latter residue. In our structure we can observe a direct contact between the oxygen of the *N*-acetyl group and Glu184 (dashed line in Figure 3c).

## 3. Materials and Methods

### 3.1. General Synthetic Methods

All reactions in organic medium were performed in standard oven-dried glassware under an inert atmosphere of nitrogen using freshly distilled solvents. Solvents and reagents were deoxygenated, when necessary, by purging with nitrogen. All reagents were used as supplied without prior purification, unless otherwise stated, and obtained from Sigma-Aldrich Chemical Co. Ltd. (St. Louis, MO, USA). Reactions were monitored by analytical thin-layer chromatography (TLC) using silica gel 60 F254 precoated plates (E. Merck, Rahway, NJ, USA), and compounds were visualized with a 254 nm UV lamp, potassium permanganate solution (1.5 g KMnO_4_, 10 g K_2_CO_3_, 1.5 mL 10% NaOH in 200 mL H_2_O), a mixture of iodine/silica gel and/or mixture of ceric ammonium molybdate solution (100 mL H_2_SO_4_, 900 mL H_2_O, 25 g (NH_4_)_6_Mo_7_O_24_H_2_O, 10 g Ce(SO_4_)_2_), and subsequent spot development by gentle warming with a heat-gun. Purifications were performed by silica gel flash column chromatography using Silica (Katy, TX, USA) gel (60 Å, 40–63 µm) with the indicated eluent. The optical rotation measurement [α]_D_ was performed at 589 nm using a Jasco (Tokyo, Japan) P2000 polarimeter. NMR spectroscopy was used to record ^1^H NMR and ^13^C NMR spectra at 300 or 600 MHz and at 75 or 150 MHz, respectively, on Bruker (Billerica, MA, USA) (300 MHz) and Bruker Avance III HD 600 MHz spectrometers. Proton and carbon chemical shifts (δ) are reported in ppm relative to the chemical shift of residual CDCl_3_ (in ^1^H 7.26 ppm, in ^13^C 77.16 ppm), CD_3_OD (in ^1^H, 3.31 ppm and in ^13^C, 49.0 ppm), D_2_O (in ^1^H, 4.79 ppm), which were set respectively. 2D homonuclear COrrelationSpectroscopY^1^H-^1^H (COSY), ^1^H-^13^C Heteronuclear Single Quantum Coherence (HSQC), and Heteronuclear Multiple bond Correlation (HMBC) experiments were used to confirm NMR peak assignments. Coupling constants (J) are reported in Hertz (Hz), and the following abbreviations are used for peak multiplicities: singlet (s), broad singlet (brs), doublet (d), doublet of doublets (dd), doublet of doublets of doublets (ddd), triplet (t), doublet of triplets (dt), triplet of doublets (td), triplet of triplets (tt), multiplet (m). High-resolution mass spectrometry (HRMS) data were measured with a LC-MS-TOF (Liquid Chromatography-Mass Spectrometry-Time of Flight; Agilent Technologies, Santa Clara, CA, USA) in positive and/or negative electrospray mode (s) at the analytical platform of UQAM. Methyl lactoside (**1**, methyl 4-*O*-β-D-galactopyranosyl-β-D-glucopyranoside) was obtained from Sigma-Aldrich (Markham, ON, Canada).

### 3.2. General Synthetic Procedure A: Preparation of 3′-O-Sulfated Lactosides

A mixture of deacetylated lactosides (1 eq.) and dibutyltin oxide (Bu_2_SnO, 1.08 eq.) in MeOH (4 mL per 0.1 mmol of the lactoside) was stirred at 80 °C for 4 h under nitrogen atmosphere. The solution was then concentrated, and sulfur trioxide-triethylamine complex (Et_3_N.SO_3_) (1.2 eq.) and dry DMF (4 mL per 0.1 mmol of the lactoside) were added. After stirring at 60 °C for 17 h, the reaction was quenched with methanol and the reaction mixture was concentrated in vacuo. The residue was purified through a classical column chromatography to give desired compound.

### 3.3. General Synthetic Procedure B: Zemplén Transesterification Reaction

To a solution of lactoside (1 eq.) in dry methanol (2 mL per 0.1 mmol of the lactoside) was added a solution of sodium methoxide (25% in MeOH, 0.5 eq.). After stirring at room temperature for 3 h, the basic media was neutralized by addition of ion-exchange resin (Amberlite IR 120 H^+^). The reaction mixture was filtered through a pad of celite and concentrated in vacuo to afford the de-O-acetylated lactosides.

### 3.4. General Synthetic Procedure C: Protection of Primary Alcohol with Tert-Butyldiphenylsilyl Ether (TBDPS)

To a solution of lactoside (1 eq.) in pyridine (2 mL per 0.1 mmol of lactoside) was added TBDPSCl (1.5 eq. per primary alcohol) at room temperature under nitrogen atmosphere. After 8 h of stirring the reaction was quenched with methanol, the reaction mixture was co-evaporated with toluene under vacuum, and the residue was purified using column chromatography.

### 3.5. General Synthetic Procedure D: Deprotection of Tert-Butyldiphenylsilyl Ether (TBDPS)

To a solution of silyl ether lactoside (1 eq.) in pyridine (10 mL per 0.1 mmol of the lactoside) was added HF·Py (70% in pyridine, 0.1 mL per 0.1 mmol) at 0 °C and the resulted mixture was stirred at room temperature for 24 h. The reaction was quenched with solid NaHCO_3_ at 0 °C, the solvent was co-evaporated with toluene, and the residue was purified using column chromatography.

#### 3.5.1. Propargyl (2,3,4,6-Tetra-O-Acetyl-β-d-Galactopyranosyl)-(1→4)-2,3,6-Tri-O-Acetyl-β-d-Glucopyranoside (**3**) [29]

A solution of compound **2** (1.195 g, 1.76 mmol, 1 eq.), propargyl alcohol (0.140 mL, 2.336 mmol, 1.32 eq.), and pulverized activated 4Å MS (1.05 g) in dry DCM (15 mL) was stirred under nitrogen at room temperature for 0.5 h. The reaction mixture was cooled to 0 °C followed by addition of BF_3_·OEt_2_ (0.32 mL, 2.592 mmol, 1.47 eq.). After stirring of 9 h, the reaction was quenched by addition of triethylamine (0.5 mL). The reaction mixture was filtered through a pad of celite and concentrated in vacuo. After chromatographic purification, compound **3** (937 mg, 1.39 mmol, 79%) was obtained as a white solid. R_f_ = 0.24, (EtOAc/Hex: 1/1). [α] _D_^23^ = −28.9 (c 0.6, CHCl_3_). ^1^H NMR (300 MHz, CDCl_3_): δ (ppm) 5.33 (dd, J = 3.4, 1.2 Hz, 1H), 5.22 (t, J = 9.2 Hz, 1H), 5.10 (dd, J = 10.4, 7.8 Hz, 1H), 4.99–4.84 (m, 2H), 4.73 (d, J = 7.9 Hz, 1H), 4.55–4.41 (m, 2H), 4.32 (d, J = 2.4 Hz, 2H), 4.18–4.00 (m, 3H), 3.92–3.74 (m, 2H), 3.62 (ddd, J = 9.9, 4.9, 2.1 Hz, 1H), 2.45 (t, J = 2.4 Hz, 1H), 2.14 (s, 3H), 2.11 (s, 3H), 2.05 (s, 3H), 2.04 (s, 3H), 2.03 (s, 6H), 1.95 (s, 3H) (Appendix A); ^13^C NMR (75 MHz, CDCl_3_): δ (ppm) 170.5, 170.3, 170.2, 169.8, 169.2, 101.2, 98.0, 78.2, 76.2, 75.6, 71.4, 71.1, 70.8, 69.2, 66.7, 61.9, 60.9, 56.0, 21.0, 20.9, 20.8, 20.8, 20.6 (Appendix A). HRMS: m/z calcd. for C_29_H_38_O_18_ [M + NH_4_]^+^, 692.2396; found 692.2389 [M + NH_4_]^+^.

#### 3.5.2. Propargyl (β-d-Galactopyranosyl)-(1→4)-β-d-Glucopyranoside (**4**) [29]

Following the general procedure B, compound **4** (283 mg, 0.746 mmol, quant.) was obtained as a white solid. [α] _D_^23^ = −33.9 (c 0.5, D_2_O). ^1^H NMR (300 MHz, CDCl_3_): δ (ppm) 4.65 (d, J = 7.9 Hz, 1H), 4.47–4.42 (m, 2H), 4.09–3.86 (m, 2H), 3.82–3.44 (m, 8H) 3.33 (t, J = 6.3 Hz, 1H) (Appendix A); ^13^C NMR (75 MHz, CDCl_3_): δ (ppm) 102.9, 100.4, 78.2, 75.3, 74.8¸74.3, 72.6, 72.5, 70.9, 68.5, 61.0, 60.0, 56.6 (Appendix A). HRMS: m/z calcd. for C_15_H_24_O_11_ [M + Na]^+^, 403.1211; found 403.1203 [M + Na]^+^.

#### 3.5.3. 4-Nitrophenyl (2,3,4,6-Tetra-O-Acetyl-β-d-Galactopyranosyl)-(1→4)-2,3,6-Tri-O-Acetyl-β-d-Glucopyranoside (**5**) [15]

To a solution of compound **2** (3.054 g, 4.5 mmol, 1 eq.) in dichloromethane (50 mL) was added HBr (33% in AcOH, 12 mL) over 0.5 h at 0 °C. After 3.5 h of stirring the yellow mixture was poured into ice-cold water (80 mL), extracted with DCM (3 × 90 mL), washed with water (5 × 60 mL), saturated solution of NaHCO_3_ (until pH = 7), dried over Na_2_SO_4_ and concentrated in vacuo. The residue was dissolved in DCM (45 mL) and mixed with a separately prepared solution of 4-nitrophenol (1.377 g, 9.9 mmol 2 eq.), tetrabutylammonium hydrogen sulfate (TBAHS, 1.528 g, 4.5 mmol 1 eq.) in NaOH (1 M, 45 mL). The mixture was stirred at room temperature for 3 h and then diluted in DCM (100 mL) washed successively with NaOH (1 M, 2 × 50 mL), saturated solution of ammonium chloride (2 × 50 mL) and brine solution (50 mL), dried over Na_2_SO_4_ and concentrated under reduced pressure. After chromatographic purification, compound **5** (2.08 g, 2.745 mmol, 61%) was obtained as a white solid. R_f_ = 0.56, (EtOAc/Hex: 6/4). [α] _D_^23^ = −28.9 (c 0.6, CHCl_3_). ^1^H NMR (300 MHz, CDCl_3_): δ (ppm) 8.18 (d, J = 9.0 Hz, 2H), 7.04 (d, J = 9.2 Hz, 1H), 5.34 (d, J = 3.1 Hz, 1H), 5.32–5.24 (m, 1H), 5.22–5.05 (m, 3H), 4.96 (dd, J = 10.4, 3.3 Hz, 1H), 4.55–4.44 (m, 2H), 4.19–4.00 (m, 3H), 3.96–3.79 (m, 3H), 2.13 (s, 3H), 2.06 (s, 3H), 2.05 (s, 6H), 2.04 (s, 6H), 1.95 (s, 3H) (Appendix A); ^13^C NMR (75 MHz, CDCl_3_): δ (ppm) 170.4, 170.2, 170.2, 170.1, 169.7, 169.6, 169.2, 161.3, 143.3, 125.8, 116.7, 101.2, 97.8, 76.0, 73.2, 72.7, 71.3, 71.0, 70.9, 69.2, 66.7, 61.9, 60.9, 20.8, 20.8, 20.7, 20.6 (Appendix A). HRMS: m/z calcd. for C_32_H_39_NO_20_ [M + Na]^+^, 780.1958; found 780.1953 [M + Na]^+^.

#### 3.5.4. 4-Nitrophenyl (β-d-Galactopyranosyl)-(1→4)-β-d-Glucopyranoside (**6**) [15]

Following the general procedure B, compound **6** (295 mg, 0.637 mmol, quant.) was obtained as a white solid. [α] _D_^23^ = −4.7 (c 0.2, H_2_O). ^1^H NMR 600 MHz, D_2_O): δ (ppm) 8.29 (d, J = 9.2 Hz, 2H), 7.27 (d, J = 9.2 Hz, 2H), 5.32 (d, J = 7.8 Hz, 1H), 4.50 (d, J = 7.8 Hz, 1H), 4.02 (d, J = 10.1 Hz, 1H), 3.95 (brs, 1H), 3.88–3.84 (m, 2H), 3.81–3.75 (m, 5H), 3.73–3.65 (m, 2H), 3.58 (t, J = 8.8 Hz, 1H) (Appendix A); ^13^C NMR (150 MHz, DMSO-d6): δ (ppm) 162.3, 141.7, 125.7, 116.6, 103.8, 99.3, 79.9, 75.6, 75.1, 74.7, 73.3, 72.9, 70.6, 68.1, 60.4, 59.9 (Appendix A). HRMS: m/z calcd. for C_18_H_25_NO_13_ [M + Na]^+^, 486.1218; found 486.1206 [M + Na]^+^.

#### 3.5.5. Propargyl (6-O-Tert-Butyldiphenylsilyl-β-d-Galactopyranosyl)-(1→4)-6-O-Tert-Butyldiphenylsilyl-β-d-Glucopyranoside (**7**)

Following the general procedure C, compound **7** (1.711 g, 1.996 mmol, 95%) was obtained as a white solid. R_f_ = 0.2, (EtOAc/MeOH: 9.5/0.5). [α] _D_^22^ = −23.8 (c 0.2, MeOH). ^1^H NMR (600 MHz, CDCl_3_): δ (ppm) 7.83–7.76 (m, 4H), 7.72–7.66 (m, 4H), 7.51–7.32 (m, 12H), 4.61 (d, J = 7.7 Hz, 1H), 4.49 (d, J = 7.8 Hz, 1H), 4.43 (dd, J = 15.5, 2.5 Hz, 1H), 4.32 (dd, J = 15.5, 2.4 Hz, 1H), 4.25 (dd, J = 11.6, 3.0 Hz, 1H), 4.02–3.81 (m, 5H), 3.66–3.52 (m, 3H), 3.48–3.42 (m, 2H), 3.35–3.32 (m, 1H), 2.88 (t, J = 2.4 Hz, 1H), 1.06 (s, 9H), 1.03 (s, 9H); ^13^C NMR (150 MHz, CDCl_3_): δ (ppm) 137.0, 136.6, 136.6, 134.8, 134.3, 134.3, 134.1, 130.8, 130.8, 130.7, 130.7, 128.8, 128.8, 128.6, 104.4, 101.6, 79.8, 78.3, 76.7, 76.4, 76.4, 75.8, 75.0, 74.7, 72.3, 69.6, 63.5, 63.0, 56.2, 27.4, 20.2, 19.9. HRMS: m/z calcd. for C_47_H_60_NO_11_Si_2_ [M + Na]^+^, 879.3566; found 879.3569 [M + Na]^+^.

#### 3.5.6. 4-Nitrophenyl (6-O-Tert-Butyldiphenylsilyl-β-d-Galactopyranosyl)-(1→4)-6-O-Tert-Butyldiphenylsilyl-β-d-Glucopyranoside (**8**)

Following the general procedure C, compound **8** (739 mg, 0.786 mmol, 93%) was obtained as a white solid. R_f_ = 0.66, (EtOAc/MeOH: 9.9/0.1). [α] _D_^21^ = −11.67 (c 1.1, MeOH). ^1^H NMR (300 MHz, CDCl_3_): δ (ppm) 8.06 (d, J = 8.9 Hz, 2H), 7.72–7.54 (m, 8H), 7.41–7.19 (m, 10H), 7.08 (t, J = 7.5 Hz, 2H), 7.02 (d, J = 9.0 Hz, 2H), 4.95 (d, J = 7.2 Hz, 1H), 4.52 (d, J = 7.5 Hz, 1H), 4.04 (d, J = 12.2 Hz, 1H), 3.98 (d, J = 3.0 Hz, 1H), 3.95–3.67 (m, 7H), 3.67–3.21 (m, 3H), 1.03 (s, 9H), 1.01 (s, 9H) (Appendix A); ^13^C NMR (75 MHz, CDCl_3_): δ (ppm) 161.9, 142.8, 135.8, 135.6, 135.5, 133.4, 132.8, 132.4, 130.1, 129.9, 128.0, 128.0, 127.9, 127.6, 125.8, 116.7, 103.2, 99.6, 77.8, 75.7, 75.3, 74.5, 73.9, 73.7, 71.4, 68.5, 62.3, 62.2, 26.9, 26.8, 19.4, 19.2 (Appendix A). HRMS: m/z calcd. for C_50_H_61_NO_13_Si_2_ [M + Na]^+^, 962.3574; found 962.3565 [M + Na]^+^.

#### 3.5.7. Propargyl (3-O-Sulfo-6-O-Tert-Butyldiphenylsilyl-β-d-Galactopyranosyl)-(1→4)-6-O-Tert-Butyldiphenylsilyl-β-d-Glucopyranoside Sodium Salt (**9**)

Following the general procedure A, compound **9** (419 mg, 0.436 mmol, 88%) was obtained as a white solid. R_f_ = 0.15, (EtOAc/MeOH: 9/1). [α] _D_^21^ = −21.4 (c 1.0, MeOH). ^1^H NMR (600 MHz, CD_3_OD): δ (ppm) 7.81–7.75 (m, 4H), 7.75–7.62 (m, 4H), 7.56–7.35 (m, 12H), 4.73 (d, J = 7.8 Hz, 1H), 4.48 (d, J = 7.8 Hz, 1H), 4.44 (d, J = 15.5 Hz, 1H), 4.36 (d, J = 2.9 Hz, 1H), 4.32 (d, J = 15.5 Hz, 1H), 4.29 (dd, J = 9.7, 3.1 Hz, 1H), 4.25 (dd, J = 11.6, 2.9 Hz, 1H), 3.97–3.91 (m, 2H), 3.88 (dd, J = 10.1, 6.9 Hz, 1H), 3.83–3.75 (m, 2H), 3.63 (t, J = 6.6 Hz, 1H), 3.56 (t, J = 9.2 Hz, 1H), 3.42 (ddd, J = 9.7, 3.0, 1.6 Hz, 1H), 3.33–3.29 (m, 2H), 1.03 (s, 9H), 1.01 (s, 9H); ^13^C NMR (150 MHz, CD_3_OD): δ (ppm) 137.1, 136.7, 136.7, 136.7, 134.8, 134.4, 134.4, 130.9, 130.9, 130.9, 130.8, 129.0, 128.9, 128.8, 128.7, 104.1, 101.7, 82.1, 78.3, 76.5, 76.4, 75.9, 74.7, 71.1, 67.9, 63.2, 63.0, 56.3, 27.4, 27.4, 20.2, 20.0. HRMS: m/z calcd. for C_47_H_60_O_14_SSi_2_ [M − H]^−^, 935.3170; found 935.3161 [M − H]^−^.

#### 3.5.8. 4-Nitrophenyl (3-O-Sulfo-6-O-Tert-Butyldiphenylsilyl-β-d-Galactopyranosyl)-(1→4)-6-O-Tert-Butyldiphenylsilyl-β-d-Glucopyranoside Sodium Salt (**10**)

Following the general procedure A, compound **10** (200 mg, 0.192 mmol, 86%) was obtained as a white solid. R_f_ = 0.50, (EtOAc/MeOH: 9/1). [α] _D_^21^ = −10.47 (c 0.6, MeOH). ^1^H NMR (600 MHz, CD_3_OD): δ (ppm) 8.15 (d, J = 9.2 Hz, 2H), 7.74–7.64 (m, 6H), 7.62 (d, J = 7.1 Hz, 2H), 7.42–7.32 (m, 9H), 7.24 (d, J = 9.2 Hz, 2H), 7.17 (t, J = 7.5 Hz, 1H), 6.97 (t, J = 7.5 Hz, 2H), 5.21 (d, J = 7.7 Hz, 1H), 4.78 (d, J = 7.8 Hz, 1H), 4.41 (d, J = 2.6 Hz, 1H), 4.34 (dd, J = 9.8, 2.9 Hz, 1H), 4.25 (dd, J = 11.7, 2.5 Hz,1H), 4.02 (t, J = 9.5 Hz, 1H), 3.99 (d, J = 11.5 Hz, 1H), 3.97–3.85 (m, 2H), 3.84 (dd, J = 9.9, 6.2 Hz, 1H), 3.82–3.74 (m, 2H), 3.68 (t, J = 6.6 Hz, 1H), 3.63 (t, J = 8.0 Hz, 1H), 1.04 (s, 9H), 0.95 (s, 9H) (Appendix A); ^13^C NMR (150 MHz, CD_3_OD): δ (ppm) 163.6, 143.6, 136.8, 136.6, 136.6, 136.5, 134.6, 134.3, 134.3, 133.6, 130.9, 130.9, 130.8, 130.5, 128.9, 128.8, 128.8, 128.4, 126.6, 117.7, 104.0, 100.8, 82.0, 77.8, 76.5, 76.3, 75.4, 74.7, 70.9, 67.9, 63.2, 62.9, 27.4, 27.3, 20.2, 19.9 (Appendix A). HRMS: m/z calcd. for C_50_H_61_NO_16_SSi_2_ [M + Na]^+^, 1042.3142; found 1042.3140 [M + Na]^+^.

#### 3.5.9. Propargyl (3-O-Sulfo-β-d-Galactopyranosyl)-(1→4)-β-d-Glucopyranoside Sodium Salt (**11**)

Following the general procedure D, compound **11** (62 mg, 0.128 mmol, 81%) was obtained as a white solid. R_f_ = 0.30, (EtOAc/^i^PrOH/H_2_O: 5/6/2.5). [α] _D_^22^ = −28.0 (c 1.0, H_2_O). ^1^H NMR (600 MHz, D_2_O): δ (ppm) 4.68 (d, J = 8.0 Hz, 1H), 4.58 (d, J = 7.8 Hz, 1H), 4.53–4.43 (m, 2H), 4.34 (dd, J = 9.9, 3.2 Hz, 1H), 4.30 (d, J = 3.3 Hz, 1H), 4.00 (dd, J = 12.4, 2.2 Hz, 1H), 3.93–3.75 (m, 4H), 3.74–3.65 (m, 3H), 3.66–3.60 (m, 1H), 3.40–3.28 (m, 1H), 2.93 (t, J = 2.4 Hz, 1H); ^13^C NMR (150 MHz, D_2_O): δ (ppm) 102.6, 100.4, 80.0, 78.8, 78.2, 76.4, 74.9, 74.9, 74.4, 72.6, 69.1, 66.9, 61.0, 60.0, 56.6, 53.9. HRMS: m/z calcd. for C_15_H_24_O_14_S [M − H]^−^, 459.0814; found 459.0834 [M − H]^−^.

#### 3.5.10. 4-Nitrophenyl (3-O-Sulfo-β-d-Galactopyranosyl)-(1→4)-β-d-Glucopyranoside Sodium Salt (**12**)

Following the general procedure D, compound **12** (38 mg, 0.067 mmol, 74%) was obtained as a white solid. R_f_ = 0.38, (EtOAc/^i^PrOH/H_2_O: 5/6/2.5). [α] _D_^22^ = −12.7 (c 0.5, H_2_O). ^1^H NMR (600 MHz, D_2_O): δ (ppm) 8.26 (d, J = 9.2 Hz, 2H), 7.26 (d, J = 9.3 Hz, 2H), 5.31 (d, J = 7.8 Hz, 1H), 4.63 (d, J = 7.8 Hz, 1H), 4.38 (dd, J = 9.9, 3.2 Hz, 1H), 4.33 (d, J = 3.1 Hz, 1H), 4.04 (d, J = 10.6 Hz, 1H), 3.92–3.78 (m, 7H), 3.76–3.69 (m, 2H) (Appendix A); ^13^C NMR (150 MHz, D_2_O): δ (ppm) 161.7, 142.6, 126.1, 116.5, 102.6, 99.3, 80.0, 77.9, 75.1, 75.0, 74.1, 72.5, 69.1, 66.8, 60.9, 59.8 (Appendix A). HRMS: m/z calcd. for C_18_H_25_NO_16_S [M + Na]^+^, 566.0786; found 566.0784 [M + Na]^+^.

#### 3.5.11. 4-Nitrophenyl 2-N-Acetamido-2-Deoxy-β-d-Glucopyranoside (**15**) [31]

Following the general procedure B, compound **15** (0.46 g, 1.33 mmol, quant.) was obtained as a white solid without further purification. [α] _D_^22^ = −26.9 (c 1.0, CH_3_OH). ^1^H NMR (300 MHz, CD_3_OD): δ (ppm) 8.21 (d, J = 9.2 Hz, 2H), 7.18 (d, J = 9.3 Hz, 2H), 5.21 (d, J = 8.4 Hz, 1H), 4.01–3.87 (m, 2H), 3.72 (dd, J = 12.1, 5.6 Hz, 1H), 3.60 (dd, J = 10.3, 8.6 Hz, 1H), 3.56–3.46 (m, 1H), 3.49–3.37 (m, 1H), 1.98 (s, 3H); ^13^C NMR (75 MHz, CD_3_OD): δ (ppm) 173.9, 163.7, 144.0, 126.6, 117.7, 100.0, 78.5, 75.6, 71.7, 62.5, 57.1, 22.9.

#### 3.5.12. 4-Nitrophenyl 2-N-Acetamido-2-Deoxy-6-O-Tert-Butyldiphenylsilyl-β-d-Glucopyranoside (**16**)

Following the general procedure C, compound **16** (0.73 g, 1.25 mmol, 91%) was obtained as a white solid. R_f_ = 0.50, (EtOAc). [α] _D_^21^ = −24.2 (c 1.0, CHCl_3_). ^1^H NMR (300 MHz, CDCl_3_): δ (ppm) 7.98 (d, J = 9.1 Hz, 2H), 7.66–7.55 (m, 4H), 7.44–7.28 (m, 2H), 7.31–7.18 (m, 4H), 7.03 (d, J = 9.2 Hz, 1H), 6.49 (brs, 1H), 5.20 (d, J = 7.8 Hz, 1H), 4.08–3.98 (m, 1H), 3.97–3.76 (m, 3H), 3.71–3.60 (m, 1H), 3.60–3.50 (m, 1H), 2.03 (s, 3H), 1.02 (s, 9H); ^13^C NMR (75 MHz, CDCl_3_): δ (ppm) 172.9, 161.8, 142.6, 135.7, 135.6, 133.1, 132.9, 130.0, 129.9, 127.8, 127.8, 125.9, 116.5, 98.0, 76.9, 75.2, 71.6, 63.9, 56.8, 26.9, 23.6, 19.4.

#### 3.5.13. 4-Nitrophenyl (2,3,4,6-Tetra-O-Acetyl-β-d-Galactopyranosyl)-(1→4)-2-Acetamido-2-Deoxy-6-O-Tert-Butyldiphenylsilyl-β-d-Glucopyranoside (**18**)

A solution of glucoside acceptor **16** (1461 mg, 2.515 mmol, 1 eq.), galactoside donor **17** (2218 mg, 4.501 mmol, 1.79 eq.), and pulverized activated 4Å MS (750 mg) in dry DCM/THF (4/1: v/v, 75 mL) was stirred under nitrogen at room temperature for 0.5 h. The reaction mixture was cooled to −35 °C followed by addition of BF_3_·OEt_2_ (0.49 mL, 3.97 mmol, 1.58 eq.). After stirring of 5 h, the reaction was quenched by addition of triethylamine (0.5 mL). The reaction mixture was filtered through a pad of celite and concentrated in vacuo. After chromatographic purification, compound **18** (871 mg, 0.956 mmol, 38%) was obtained as a white solid. R_f_ = 0.57, (EtOAc). [α] _D_^23^ = −18.3 (c 1.0, Acetone). ^1^H NMR (300 MHz, CDCl_3_): δ (ppm) 8.10 (d, J = 9.2 Hz, 2H), 7.68–7.52 (m, 4H), 7.45–7.28 (m,4H), 7.09 (t, J = 7.5 Hz, 2H), 7.05 (d, J = 9.2 Hz, 2H), 5.95 (d, J = 8.0 Hz, 1H), 5.53 (d, J = 8.1 Hz, 1H), 5.38 (dd, J = 3.4, 1.0 Hz, 1H), 5.20 (dd, J = 10.5, 8.0 Hz, 1H), 4.98 (dd, J = 10.5, 3.4 Hz, 1H), 4.71 (d, J = 8.0 Hz, 1H), 4.20–4.03 (m, 3H), 4.03–3.90 (m, 2H), 3.92–3.72 (m, 3H), 3.65 (d, J = 11.2 Hz, 1H), 2.15 (s, 3H), 2.06 (s, 3H), 2.01 (s, 3H), 1.98 (s, 3H), 1.69 (s, 3H), 1.02 (s, 9H) (Appendix A); ^13^C NMR (75 MHz, CDCl_3_): δ (ppm) 170.9, 170.5, 170.2, 170.0, 169.2, 162.0, 142.8, 135.8, 135.5, 133.3, 132.2, 130.1, 130.0, 128.0, 127.7, 125.8, 116.7, 101.2, 97.5, 79.9, 75.1, 71.5, 71.2, 70.8, 68.9, 67.0, 61.8, 61.4, 56.8, 26.9, 23.7, 20.7, 20.6, 20.6, 20.4, 19.4 (Appendix A). HRMS: m/z calcd. for C_44_H_55_N_2_O_17_Si [M + Na]^+^, 933.3084; found 933.3085 [M + Na]^+^.

#### 3.5.14. 4-Nitrophenyl (β-d-Galactopyranosyl)-(1→4)-2-Acetamido-2-Deoxy-6-O-Tert-Butyldiphenylsilyl-β-d-Glucopyranoside (**19**)

Following the general procedure B, compound **19** (340 mg, 0.472 mmol, quant.) was obtained as a white solid. [α] _D_^23^ = −23.67 (c 1.1, MeOH). ^1^H NMR (300 MHz, CD_3_OD): δ (ppm) 8.17 (d, J = 8.9 Hz, 2H), 7.73 (d, J = 8.1 Hz, 2H), 7.66 (d, J = 7.4 Hz, 2H), 7.42–7.23 (m, 4H), 7.19 (d, J = 9.0 Hz, 2H), 7.07 (t, J = 7.5 Hz, 2H), 5.29 (d, J = 8.3 Hz, 1H), 4.65 (d, J = 7.6 Hz, 1H), 4.30 (dd, J = 11.7, 3.3 Hz, 1H), 4.22–3.98 (m, 3H), 3.90–3.45 (m, 8H), 2.03 (s, 3H), 1.04 (s, 9H) (Appendix A); ^13^C NMR (75 MHz, CD_3_OD): δ (ppm) 173.8, 163.5, 143.9, 136.9, 136.7, 134.9, 134.0, 130.7, 130.6, 128.7, 128.5, 126.7, 117.7, 104.8, 99.4, 78.8, 77.4, 77.0, 75.1, 73.8, 72.5, 70.4, 63.1, 62.7, 56.6, 27.4, 23.0, 20.2 (Appendix A). HRMS: m/z calcd. for C_36_H_46_N_2_O_13_Si [M + Na]^+^, 765.2661; found 765.2646 [M + Na]^+^.

#### 3.5.15. 4-Nitrophenyl (β-d-Galactopyranosyl)-(1→4)-2-Acetamido-2-Deoxy-β-d-Glucopyranoside (**20**)

Following the general procedure D, compound **20** (71 mg, 0.140 mmol, 91%) was obtained as a white solid. R_f_ = 0.14, (EtOAc/^i^PrOH/H_2_O: 6/5/2). [α] _D_^22^ = −17.8 (c 1.0, H_2_O). ^1^H NMR (300 MHz, D_2_O): δ (ppm) 8.26 (d, J = 9.2 Hz, 2H), 7.21 (d, J = 9.3 Hz, 2H), 5.37 (d, J = 8.3 Hz, 1H), 4.54 (d, J = 7.7 Hz, 1H), 4.12–3.79 (m, 10H), 3.71 (dd, J = 10.1, 3.3 Hz, 1H), 3.59 (dd, J = 10.0, 7.6 Hz, 1H), 2.05 (s, 3H) (Appendix A); ^13^C NMR (75 MHz, D_2_O): δ (ppm) 174.9, 161.7, 142.7, 126.1, 116.5, 102.9, 98.5, 78.1, 75.4, 75.2, 72.5, 72.0, 71.0, 68.6, 61.1, 59.8, 55.5, 54.9, 22.1 (Appendix A). HRMS: m/z calcd. for C_20_H_28_N_2_O_13_ [M + Na]^+^, 527.1489; found 527.1478 [M + Na]^+^.

#### 3.5.16. 4-Nitrophenyl (6-O-Tert-Butyldiphenylsilyl-β-d-Galactopyranosyl)-(1→4)-2-Acetamido-2-Deoxy-6-O-Tert-Butyldiphenylsilyl-β-d-Glucopyranoside (**21**)

Following the general procedure C, compound **21** (253 mg, 0.258 mmol, 94%) was obtained as a white solid. R_f_ = 0.34, (EtOAc). [α] _D_^23^ = −7.45 (c 0.4, MeOH). ^1^H NMR (300 MHz, CD_3_OD): δ (ppm) 8.14 (d, J = 9.3 Hz, 2H), 7.74–7.57 (m, 8H), 7.46–7.27 (m, 9H), 7.22 (t, J = 7.5 Hz, 1H), 7.16 (d, J = 9.3 Hz, 2H), 7.03 (t, J = 7.6 Hz, 2H), 5.27 (d, J = 8.4 Hz, 1H), 4.58 (d, J = 7.7 Hz, 1H), 4.23 (dd, J = 11.5, 3.6 Hz, 1H), 4.09–4.00 (m, 2H), 3.99–3.78 (m, 5H), 3.74 (d, J = 9.7 Hz, 1H), 3.66–3.55 (m, 2H), 3.47 (dd, J = 9.7, 3.2 Hz, 1H), 1.96 (s, 3H), 1.05 (s, 9H), 0.99 (s, 9H) (Appendix A); ^13^C NMR (75 MHz, CD_3_OD): δ (ppm) 173.7, 163.5, 143.9, 136.9, 136.7, 136.7, 134.8, 134.5, 134.4, 133.9, 130.9, 130.7, 130.6, 128.9, 128.9, 128.8, 128.5, 126.7, 117.7, 105.0, 99.4, 79.5, 77.0, 76.9, 75.1, 73.5, 72.4, 69.8, 63.8, 63.2, 56.7, 27.4, 27.4, 23.0, 20.2, 20.0 (Appendix A). HRMS: m/z calcd. for C_52_H_64_N_2_O_13_Si_2_ [M + Na]^+^, 1003.3839; found 1003.3835 [M + Na]^+^.

#### 3.5.17. 4-Nitrophenyl (3-O-Sulfo-6-O-Tert-Butyldiphenylsilyl-β-d-Galactopyranosyl)-(1→4)-2-Acetamido-2-Deoxy-6-O-Tert-Butyldiphenylsilyl-β-d-Glucopyranosidesodiumsalt (**22**)

Following the general procedure A, compound **22** (119 mg, 0.11 mmol, 83%) was obtained as a white solid. R_f_ = 0.09, (EtOAc/MeOH: 9/1). [α] _D_^22^ = −23.7 (c 0.011, MeOH). ^1^H NMR (300 MHz, CD_3_OD): δ (ppm) 8.16 (d, J = 9.3 Hz, 2H), 7.81–7.50 (m, 8H), 7.49–7.25 (m, 9H), 7.26–7.10 (m, 3H), 6.98 (t, J = 7.6 Hz, 2H), 5.36 (d, J = 8.4 Hz, 1H), 4.74 (d, J = 7.8 Hz, 1H), 4.37–4.29 (m, 2H), 4.22 (dd, J = 11.7, 3.1 Hz, 1H), 4.13 (dd, J = 10.3, 8.4 Hz, 1H), 4.05–3.77 (m, 7H), 3.68 (t, J = 6.6 Hz, 1H), 1.97 (s, 3H), 1.04 (s, 9H), 0.97 (s, 9H) (Appendix A); ^13^C NMR (75 MHz, CD_3_OD): δ (ppm) 173.7, 163.5, 143.8, 136.9, 136.7, 136.5, 134.6, 134.3, 133.7, 130.9, 130.8, 130.6, 128.9, 128.9, 128.8, 128.4, 126.6, 117.7, 104.4, 99.4, 82.0, 79.0, 76.6, 76.4, 73.5, 71.0, 68.0, 63.3, 62.9, 56.4, 27.4, 27.3, 23.0, 20.2, 20.0 (Appendix A). HRMS: m/z calcd. for C_52_H_64_N_2_O_16_SSi_2_ [M + Na]^+^, 1083.3407; found 1083.3358 [M + Na]^+^.

#### 3.5.18. 4-Nitrophenyl (3-O-Sulfo-β-d-Galactopyranosyl)-(1→4)-2-Acetamido-2-Deoxy-β-d-Glucopyranoside Sodium Salt (**23**)

Following the general procedure D, compound **23** (43 mg, 0.071 mmol, 72%) was obtained as a white solid. R_f_ = 0.54, (EtOAc/^i^PrOH/H_2_O: 6/5/3). [α] _D_^23^ = −10.8 (c 0.3, H_2_O). ^1^H NMR (600 MHz, D_2_O): δ (ppm) 8.35 (d, J = 9.0 Hz, 2H), 7.30 (d, J = 9.1 Hz, 2H), 5.48 (d, J = 8.4 Hz, 1H), 4.73 (d, J = 7.9 Hz, 1H), 4.47 (dd, J = 10.0, 3.3 Hz, 1H), 4.40 (d, J = 3.3 Hz, 1H), 4.19 (t, J = 9.0 Hz, 1H), 4.12 (d, J = 11.8 Hz, 1H), 4.06–3.95 (m, 3H), 3.92–3.84 (m, 4H), 3.81 (dd, J = 9.9, 8.0 Hz, 1H), 2.12 (s, 3H) (Appendix A); ^13^C NMR (75 MHz, D_2_O): δ (ppm) 175.8, 161.7, 142.7, 126.3, 116.7, 102.6, 98.6, 79.9, 77.9, 75.1, 75.0, 72.0, 69.2, 66.9, 61.0, 59.9, 54.9, 22.3 (Appendix A). HRMS: m/z calcd. for C_20_H_29_N_2_O_16_S [M + H]^+^, 585.1232; found 585.1222 [M + H]^+^.

#### 3.5.19. Propargyl (β-d-Galactopyranosyl)-(1→4)-2-Acetamido-2-Deoxy-6-O-Tert-Butyldiphenylsilyl-β-d-Glucopyranoside (**26**)

Following the general procedure B, compound **26** (326 mg, 0.495 mmol, quant.) was obtained as a white solid. [α] _D_^21^ = −12.8 (c 2.0, MeOH). ^1^H NMR (300 MHz, CD_3_OD): δ (ppm) 7.65 (d, J = 7.2 Hz, 4H), 7.38–7.26 (m, 6H), 4.59 (d, J = 7.4 Hz, 1H), 4.57 (d, J = 6.9 Hz, 1H), 4.29–4.09 (m, 3H), 3.99–3.86 (m, 2H), 3.82–3.60 (m, 5H), 3.57–3.39 (m, 4H), 3.27 (brs, 1H), 1.99 (s, 3H), 0.95 (s, 9H) (Appendix A); ^13^C NMR (75 MHz, CD_3_OD): δ (ppm) 174.6, 136.7, 136.5, 134.5, 133.9, 130.9, 128.8, 128.6, 103.9, 99.7, 78.0, 76.9, 76.3, 74.3, 73.7, 72.1, 69.9, 62.9, 62.3, 56.2, 56.2, 27.3, 23.1, 20.0 (Appendix A). HRMS: m/z calcd. for C_33_H_45_NO_11_Si [M + Na]^+^, 682.2654; found 682.2662 [M + Na]^+^.

#### 3.5.20. Propargyl (β-d-Galactopyranosyl)-(1→4)-2-Acetamido-2-Deoxy-β-d-Glucopyranoside (**27**) [19]

To a solution of compound **25** (200 mg, 0.241 mmol) in dry THF (5 mL) was added TBAF (1M in THF, 0.5 mL). The reaction mixture was stirred at room temperature for 14 h, then the reaction mixture was concentrated in vacuo, the residue was dissolved in MeOH/THF (1:1 v/v, 5 mL), and NaOMe (25% in MeOH, 0.05 mL) was added. After 3 h of stirring, the basic media was neutralized using ion-exchange resin Amberlite 120 H^+^. After chromatographic purification, compound **27** (89 mg, 0.241 mmol, 88%) was obtained as a white solid. R_f_ = 0.44 (EtOAc/^i^PrOH/H_2_O: 6/5/2). [α] _D_^20^ = −26.3 (c 1.0, H_2_O). ^1^H NMR (300 MHz, D_2_O): δ (ppm) 4.49 (d, J = 7.7 Hz, 1H), 4.43 (d, J = 1.9 Hz, 2H), 4.01 (d, J = 11.3 Hz, 1H), 3.91 (dd, J = 21.3, 3.8 Hz, 2H), 3.91 (dd, J = 21.3, 3.8 Hz, 2H), 3.89–3.51 (m, 9H), 3.61–3.45 (m, 1H), 2.94 (brs, 1H), 2.06 (s, 3H) (Appendix A); ^13^C NMR (75 MHz, D_2_O): δ (ppm) 174.7, 102.9, 99.3, 78.8, 78.3, 76.2, 75.3, 75.3, 72.5, 71.0, 68.5, 61.0, 60.0, 56.7, 54.9, 22.2 (Appendix A). HRMS: m/z calcd. for C_17_H_27_NO_11_ [M + Na]^+^, 444.1482; found 444.1477 [M + Na]^+^.

#### 3.5.21. Propargyl (6-O-Tert-Butyldiphenylsilyl-β-d-Galactopyranosyl)-(1→4)-2-Acetamido-2-Deoxy-6-O-Tert-Butyldiphenylsilyl-β-d-Glucopyranoside (**28**)

Following the general procedure C, compound **28** (249 mg, 0.277 mmol, 96%) was obtained as a white solid. R_f_ = 0.54, (EtOAc/MeOH: 9.5/0.5). [α] _D_^21^ = −15.8 (c 4.0, MeOH). ^1^H NMR (300 MHz, CD_3_OD): δ (ppm) 7.80–7.76 (m, 4H), 7.70–7.65 (m, 4H), 7.42–7.28 (m, 12H), 4.64 (d, J = 8.0 Hz, 1H), 4.59 (d, J = 7.7 Hz, 1H), 4.43–4.15 (m, 3H), 3.97 (d, J = 12.1 Hz, 1H), 3.92–3.83 (m, 4H), 3.81–3.68 (m, 2H), 3.67–3.57 (m, 2H), 3.47 (dd, J = 9.7, 3.1 Hz, 2H), 2.84 (t, J = 2.4 Hz, 1H), 1.96 (s, 3H), 1.04 (s, 9H), 1.03 (s, 9H) (Appendix A); ^13^C NMR (75 MHz, CD_3_OD): δ (ppm) 173.5, 137.0, 136.7, 136.6, 134.8, 134.3, 134.3, 134.1, 130.9, 130.9, 130.8, 130.7, 128.8, 128.8, 128.6, 104.7, 100.0, 79.8, 79.5, 76.7, 76.5, 76.4, 75.0, 73.6, 72.3, 69.6, 63.6, 63.2, 56.5, 56.0, 27.4, 27.4, 23.0, 20.2, 19.9 (Appendix A). HRMS: m/z calcd. for C_49_H_63_NO_11_Si_2_ [M + Na]^+^, 920.3832; found 920.3830 [M + Na]^+^.

#### 3.5.22. Propargyl (3-O-Sulfo-6-O-Tert-Butyldiphenylsilyl-β-d-Galactopyranosyl)-(1→4)-2-Acetamido-2-Deoxy-6-O-Tert-Butyldiphenylsilyl-β-d-Glucopyranoside Sodium Salt (**29**)

Following the general procedure A, compound **29** (161 mg, 0.161 mmol, 90%) was obtained as a white solid. R_f_ = 0.08, (EtOAc/MeOH: 9/1). [α] _D_^21^ = −10.8 (c 0.5, MeOH). ^1^H NMR (300 MHz, CD_3_OD): δ (ppm) 7.81–7.75 (m, 4H), 7.69–7.63 (m, 4H), 7.50–7.31 (m, 12H), 4.72 (d, J = 7.7 Hz, 1H), 4.63 (d, J = 8.2 Hz, 1H), 4.42–4.19 (m, 5H), 3.96–3.62 (m, 8H), 3.46 (d, J = 9.0 Hz, 1H), 2.87 (t, J = 2.7 Hz, 1H), 1.96 (s, 3H), 1.03 (s, 9H), 1.02 (s, 9H) (Appendix A); ^13^C NMR (75 MHz, CD_3_OD): δ (ppm) 173.6, 137.0, 136.7, 136.6, 134.7, 134.3, 134.3, 134.1, 130.9, 130.9, 130.9, 130.8, 129.0, 128.9, 128.8, 128.7, 104.1, 100.2, 82.0, 79.8, 79.1, 76.4, 76.2, 73.8, 71.0, 67.8, 63.1, 63.0, 56.3, 56.2, 55.1, 27.4, 27.3, 23.0, 20.2, 19.9 (Appendix A). HRMS: m/z calcd. for C_49_H_64_NO_14_SSi_2_ [M + H]^+^, 978.3581; found 978.3579 [M + H]^+^.

#### 3.5.23. Propargyl (3-O-Sulfo-β-d-Galactopyranosyl)-(1→4)-2-Acetamido-2-Deoxy-β-d-Glucopyranoside Sodium Salt (**30**)

Following the general procedure D, compound **30** (27 mg, 0.051 mmol, 64%) was obtained as a white solid. R_f_ = 0.27, (EtOAc/^i^PrOH/H_2_O: 5/5/2.5). [α] _D_^21^ = −11.0 (c 0.2, H_2_O). ^1^H NMR (300 MHz, D_2_O): δ (ppm) 4.61 (d, J = 7.8 Hz, 1H), 4.44 (d, J = 2.3 Hz, 2H), 4.36 (dd, J = 9.8, 3.2 Hz, 1H), 4.31 (d, J = 3.2 Hz, 1H), 4.02 (dd, J = 12.3, 2.0 Hz, 1H), 3.91–3.60 (m, 9H), 2.93 (t, J = 2.3 Hz, 1H), 2.06 (s, 3H) (Appendix A); ^13^C NMR (75 MHz, D_2_O): δ (ppm) 174.8, 102.5, 99.4, 80.0, 78.8, 78.2, 76.2, 74.9, 74.8, 72.4, 69.1. 66.8, 60.9, 59.9, 56.7, 54.8, 22.2 (Appendix A). HRMS: m/z calcd. for C_17_H_28_NO_14_S [M + H]^+^, 502.1225; found 502.1227 [M + H]^+^.

#### 3.5.24. Propargyl (3-O-[2-(1,1-Dimethylethoxy)-2-Oxoethyl]-6-O-Tert-Butyldiphenylsilyl-β-d-Galactopyranosyl)-(1→4)-2-Acetamido-2-Deoxy-6-O-Tert-Butyldiphenylsilyl-β-d-Glucopyranoside (**31**)

A mixture of compound **28** (209 mg, 0.232 mmol, 1 eq.) and dibutyltin oxide (71 mg, 279 mmol 1.2 eq.) in THF/Toluene (1/1: v/v, (4 mL) was stirred using the Dean-Stark trap at 115 °C for 4 h under nitrogen atmosphere. The solution was then concentrated, and tert-butyl bromoacetate **7** (0.3 mL, 1.97 mmol, 8.5 eq.), tetrabutylammonium bromide (TBAB, 157 mg, 0.487 mmol, 2.1 eq.) and dry THF (6 mL) were added. After stirring at 70 °C for 4 h, the reaction was cooled to room temperature and quenched with methanol, and the reaction mixture was concentrated in vacuo. The residue was purified through a classical column chromatography to give compound **31** (189 mg, 0.186 mmol, 80%) as a white solid. R_f_ = 0.27, (EtOAc). [α] _D_^22^ = −19.4 (c 1.0, MeOH). ^1^H NMR (300 MHz, CDCl_3_): δ (ppm) 7.82–7.74 (m, 4H), 7.70–7.61 (m, 4H), 7.47–7.29 (m, 12H), 6.03 (d, J = 8.3 Hz, 1H), 4.73 (d, J = 8.3 Hz, 1H), 4.52 (d, J = 7.8 Hz, 1H), 4.43–4.10 (m, 5H), 4.05–3.77 (m, 7H), 3.67 (d, J = 8.8 Hz, 1H), 3.55 (t, J = 6.6 Hz, 1H), 3.47–3.38 (m, 1H), 3.22 (dd, J = 9.5, 3.1 Hz, 1H), 2.42 (t, J = 2.4 Hz, 1H), 1.97 (s, 3H), 1.47 (s, 9H), 1.04 (s, 18H) (Appendix A); ^13^C NMR (75 MHz, CDCl_3_): δ (ppm) 171.6, 170.9, 136.0, 135.7, 135.7, 135.6, 133.8, 133.1, 133.1, 133.0, 129.9, 129.9, 129.7, 129.6, 127.9, 127.8, 127.8, 127.6, 103.9, 98.3, 84.4, 83.1, 80.1, 79.1, 75.0, 74.8, 72.0, 69.9, 67.7, 65.4, 62.3, 62.0, 56.1, 55.1, 28.1, 26.9, 23.7, 19.4, 19.2 (Appendix A). HR

MS: m/z calcd. for C_55_H_73_NO_13_Si_2_ [M + H]^+^, 1012.4693; found 1012.4679 [M + H]^+^.

#### 3.5.25. Propargyl (3-O-Carboxymethyl-β-d-Galactopyranosyl)-(1→4)-2-Acetamido-2-Deoxy-β-d-Glucopyranoside Sodium Salt (**32**)

To a solution of compound **31** (150 mg, 0.148 mmol) in DCM (2 mL) was added trifluoracetic acid (TFA, 2 mL) at 0 °C, and the reaction mixture was stirred at room temperature for 30 min. After removal of solvent in vacuo, the general procedure D was applied to give compound **32** (64 mg, 0.126 mmol, 85%) as a white solid. R_f_ = 0.15, (EtOAc/^i^PrOH/H_2_O: 6/5.5/2.5). [α] _D_^22^ = −9.0 (c 0.3, H_2_O). ^1^H NMR (600 MHz, D_2_O): δ (ppm) 4.75 (d, J = 7.7 Hz, 1H), 4.52 (d, J = 7.9 Hz, 1H), 4.42 (s, 2H), 4.10 (d, J = 3.8 Hz, 1H), 4.09 (s, 2H), 4.00 (dd, J = 12.4, 2.3 Hz, 1H), 3.86 (dd, J = 12.4, 5.0 Hz, 1H), 3.83–3.73 (m, 5H), 3.71 (dd, J = 8.1, 3.5 Hz, 1H), 3.65 (dd, J = 9.8, 7.9 Hz, 1H), 3.63–3.61 (m, 1H), 3.51 (dd, J = 9.8, 3.2 Hz, 1H), 2.05 (s, 3H) (Appendix A); ^13^C NMR (150 MHz, D_2_O): δ (ppm) 181.5, 174.8, 102.8, 99.4, 81.9, 78.2, 75.1, 74.9, 72.3, 69.9,s 68.5, 65.4, 61.1, 60.0, 56.8, 54.9, 22.3 (Appendix A). HRMS: m/z calcd. for C_19_H_29_NO_13_ [M + Na]^+^, 502.1531; found 502.1524 [M + Na]^+^.

### 3.6. Isothermal Titration Calorimetry (ITC)

ITC experiments were performed using a VP-ITC instrument from GE. Injections of 4 µL of carbohydrate solutions were added from a computer-controlled micro syringe at an interval of 4 min into the sample solution of hGal-3 (cell volume = 1.43 mL) with stirring at 350 rpm. Control experiments were performed by injecting the carbohydrates into a cell containing only buffer. The concentration range of hGal-3 was 90–130 µΜ, and the concentrations of the carbohydrates were 15–25 mM based on the dry weight molecular weights of the latter molecules. The concentration of hGal-3 was based on its molecular mass. Titrations were done at 27 °C (pH 7.2) using 20 mM PBS buffer. The experimental data were fitted to a theoretical titration curve using software supplied by the vendor, with ΔH (binding enthalpy in kcal mol^−1^) (lectin monomer units), Ka (association constant) and n (number of binding sites per monomer), as adjustable parameters. The Kd (dissociation constant) was calculated from 1/Ka. The quantity c = Ka Mt (0), where Mt (0) is the initial macromolecule concentration, is of importance in titration microcalorimetry. All experiments were performed with c values 1 < c < 500. The instrument was calibrated using the calibration kit containing ribonuclease A (RNase A) and cytidine 2′-monophosphate (2′-CMP) supplied by the manufacturer. Thermodynamic parameters were calculated from the Gibbs Free Energy equation, ΔG = ΔH − TΔS = -RT ln(Ka), where ΔG, ΔH, and ΔS are the changes in free energy, enthalpy, and entropy of binding, respectively. T is the absolute temperature and R = 1.98 cal mol^−1^1 K^−1^.

### 3.7. Crystallization of Gal-3C in Complex with ***23***

#### 3.7.1. Cloning

The hGal-3 coding sequence gene was optimized for *Escherichia coli* expression, synthesized and cloned directly into pUCIDTKan vector by the company Integrated DNA Technologies (IDT, Leuven, Belgium) and named pUCIDTKan-Gal-3. The coding sequence for the carbohydrate-recognition domain (CRD) was amplified from the vector pUCIDTKan-Gal-3 using the following primers: 5′-CCACCGGCCATATGGGCGCACCCGCTGGACC-3′ (NdeI site) and 5′-CAGGAAACAGCTATGAC-3′ (M13 reverse) and cloned into the vector pET41a (Novagen, Merck, Madrid, Spain), between NdeI and XhoI restriction sites, and named pET41-Gal-3C. This vector contains the coding sequence for the CRD from Gly108 up to the C-terminus of Gal-3 without any tag.

#### 3.7.2. Protein Expression and Purification

The vector pET41-Gal-3C was transformed into *E. coli* Tuner (DE3) cells (Novagen, Merck, Madrid, Spain). These cells were grown in 2xYT medium supplemented with kanamycin (30 mg L^−1^) at 37 °C; when the cells attained an OD_600_ of 0.6–0.8 the temperature was then dropped to 20 °C. When the temperature stabilized (approx. 15 min) the expression of the protein was induced by the addition of 0.1 mM isopropyl-β-D-thiogalactopyranoside (IPTG), and let to grow for an additional 16 h. The cells were harvested by centrifugation and resuspended in the lysis buffer containing 10 mM TRIS-HCl at pH 8.0 and 1% (v/v) Triton X-100 (Sigma-Aldrich, Merck, Madrid, Spain). The cells were lysed by sonication, and the insoluble fraction was removed by centrifugation at 45,000× *g* for 1 h; the supernatant was then loaded onto a lactosylated Sepharose 4B column. The column was washed extensively with PBS buffer containing 2 mM β-mercaptoethanol (β-ME). The protein was eluted from the column in PBS buffer containing 50 mM lactose and 2 mM β-ME. The fractions containing the protein were pooled and dialyzed against 1 L of PBS buffer containing 2 mM β-ME with four changes to remove the lactose.

#### 3.7.3. Crystallization and Data Collection

Crystallization trials were performed at 295 K using the sitting-drop vapor-diffusion method with commercial screening solutions including JBScreen Classic and Wizard I–IV (Jena Bioscience, Jena, Germany) in 96-well sitting-drop plates (Swissci MRC; Molecular Dimensions, Suffolk, England). Drops were set up by mixing equal volumes (0.2 μL) of a protein-**23** solution (8 mg mL^−1^ and 2 mM, respectively) and reservoir solution using a Cartesian Honeybee System (Genomic Solutions, Irvine, CA, USA) nano-dispenser robot and equilibrated against 50 μL of reservoir solution. Crystals of the protein in complex with **23** appeared after one week in 0.1 M TRIS-HCl at pH 8.5 containing 28% PEG 6000. New crystals were grown in 0.1 M TRIS-HCl at pH 8.5 containing 22–36% PEG 6000 at a protein concentration of 16 mg mL^−1^ in the presence of 2 mM of **23**.

Crystals were harvested, for data collection, in 0.1 M TRIS-HCl at pH 8.5 containing 36% PEG 6000, which is a cryo-solution, and flash-cooled in liquid nitrogen. X-Ray data collection experiments were performed at the ALBA Synchrotron (Cerdanyola del Vallès, Spain) BL13 XALOC beamline. Data were indexed and integrated, scaled and merged using the software AutoProc (https://www.globalphasing.com/autoproc/) [36] and Staraniso (https://staraniso.globalphasing.org/cgi-bin/staraniso.cgi) [37] from Globalphasing using XDS [38] and the programs POINTLESS (https://www.ccp4.ac.uk/html/pointless.html) [39], AIMLESS (https://www.ccp4.ac.uk/html/aimless.html) [40] from the CCP4 suite (https://www.ccp4.ac.uk/) [41].

#### 3.7.4. Structure Determination

The structure was solved by molecular replacement using the CRD of the previously reported Gal-3 structure [42] (PDB: 6FOF) with Phaser [43]. The initial model was first refined using Phenix [44] and alternating manual building with Coot [45]. The final model was obtained by repetitive cycles of refinement; solvent molecules were added automatically and inspected visually for chemically plausible positions. The inhibitor molecule was added manually. The stereochemical quality of the model was assessed with MolProbity [46]. The structural figures were generated using the Pymol program (http://www.pymol.org). Data processing and refinement statistics are listed in Table 2.

## 4. Conclusions

This work described the syntheses of three series of lactose and N-acetyl-lactosamine-based analogs cooperatively modified at three key positions previously known to independently improve binding to human galectin-3 (hGal-3). ITC experiments showed that the best ligand belonged to the LacNAc family with an anionic sulfate group installed at the 3′-position. Interestingly, the nature of the aglycon had little influence on the overall affinities within the Lac and LacNAc series, an observation also corroborated by the X-ray data which showed the aglycon protruding outside the active CRD-binding pocket of hGal-3. The trisaccharide containing the sialic acid moiety at the 3′-position of the LacNAc analog can also be accommodated within the active site but did not show further improvement in binding affinity. The three best candidates all contained a 3′-sulfate group. Altogether, our data substantiate the important role of 3’-sulfated LacNAc as powerful antagonists for hGal-3. These results are relevant in the context of the search for more selective and effective inhibitors of Gal-3 in the treatment of cancer, inflammation and fibrosis. These newly designed inhibitors should be compared with other small molecule inhibitors, large polysaccharide antagonists, peptidomimetics and biological agents using in vitro and in vivo biological assays. Ongoing work is being performed on Gal-1 to further explore if selectivity amongst the galectin families is effective in the selection of improved ligands.

## Data Availability

The structure of the Gal3C in complex with **23** has been deposited in the Protein Data Bank with the access code 8BZ3.

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
