# Peer review of "Selectively Modified Lactose and *N*-Acetyllactosamine Analogs at Three Key Positions to Afford Effective Galectin-3 Ligands [Author-notes fn1-ijms-24-03718]"

_ijms, 2023, doi:10.3390/ijms24043718_

Round 1

Reviewer 1 Report

The manuscript by Abdullayev et al. describes a study on the effect of multiple substitutions in lactose and lactosamine on their affinity for Gal-3. All compounds tested are synthesized specifically for the study and each compound’s interaction with Gal-3 was investigated using ITC. The crystal structure of one of the high-affinity binder with the CRD of Gal-3 was also determined. The work is thorough, rigorous, and of high quality. Publication is recommended. However, inclusion of additional clarifications would enhance the readability and informativeness of the manuscript. In particular:

11)      The manuscript title mentioned “three key positions”. However, there is no explicit mention of what the three key positions are? Are they 3-O of galactose, C2 of glucose/glucosamine, and the anomeric carbon of glucose?

22)      Although the 3-O-sulfation enhanced the affinity of the compounds, the Kds are still significantly larger than those for aryl thioglycoside compounds (e. g. Verteramo et al. J. Am. Chem. Soc. 2019, 141, 5, 2012–2026). What is the advantage of these compounds compared to aryl thioglycosides? Is it specificity?

33)      Although Table 1 is already large, inclusion of information on what substitutions each compound has at the three positions would make the table more informative. Without that information, the reader is forced to refer back to previous figures to correlate substitutions with binding affinity.

44)      It’s difficult to see any correlation between structure and the enthalpic/entropic contributions to binding, but lactosamines with no negative charges near the 3-O of galactose seem to have large and negative entropic contributions to binding. Is there any physical explanations for this?

Minor typographical errors:

Line 18, “overlay” should be “overly” or just “over”.

Line 65, “sulfate” should be “sulfated”

Scheme 1, for compound 8, R2 should be R1

Line 211, “trough” should be “through”

Author Response

The manuscript title mentioned “three key positions”. However, there is no explicit mention of what the three key positions are? Are they 3-O of galactose, C2 of glucose/glucosamine, and the anomeric carbon of glucose?

Reply:

The suggestion made by the reviewer, although relevant, would greatly complicate the title. This information was already included into the abstract.

Although the 3-O-sulfation enhanced the affinity of the compounds, the Kds are still significantly larger than those for aryl thioglycoside compounds (e. g. Verteramo et al. J. Am. Chem. Soc. 2019, 141, 5, 2012–2026). What is the advantage of these compounds compared to aryl thioglycosides? Is it specificity?

Reply:

The cited study provides dramatically different chemical entities. Indeed, the thioglycosides are sligthly better than our best compounds in terms of Kd, however, the entropy lost in the cited paper are a lot worst than in our case. This is likely due to the fact that these authors do not use a disccharide, but rather,  highly flexible aglycons having opposite stereogenic centers. However, we found the related discussion in entropy-enthalpy compensation very valuable. For this reason we changed our ref 22 by the one suggested by the reviewer (Verteramo et al.). Actually, the original ref 22 and the new one originate from the same group (A. Leffler), hence, without formal consequence. In addition,  we added comments at the end of section 2.2 (ITC) to include the reviewer"s concern.

Although Table 1 is already large, inclusion of information on what substitutions each compound has at the three positions would make the table more informative. Without that information, the reader is forced to refer back to previous figures to correlate substitutions with binding affinity.

Reply:

This is a valuable suggestion and we did include the corresponding chemical structures. However, the editorial staff may wish to improve the Table 1 to better fit the document and make the pasted structures more uniform in size.

It’s difficult to see any correlation between structure and the enthalpic/entropic contributions to binding, but lactosamines with no negative charges near the 3-O of galactose seem to have large and negative entropic contributions to binding. Is there any physical explanations for this?

Reply:

Indeed, this is the case. The results are likely associated with water solvation, as explained above and mentioned at the end of section 2.2.

Minor typographical errors:

Line 18, “overlay” should be “overly” or just “over”. We changed for "overly"

Line 65, “sulfate” should be “sulfated”. We changed for 'sulfated'

Scheme 1, for compound 8, R2 should be R1; We changed it to R1; note that we also changed scheme 3 because cpd 25 should be 'OR' rather than just 'R'

Line 211, “trough” should be “through”: We corrected it

Reviewer 2 Report

The authors described the modification of lactose at C1, C2 and O-3' positions to find effective Galectin-3 ligands. Firstly, the authors sythesized lactose analogs (4, 6, 11 and 12) and Lac-NAc analogs (20, 23, 27, 30, 32 and 33). Then their binding affinities and stoichiometry with hGal-3 were measured by ITC. In addition, the structure of Gal-3C in complex with compound 23 was obtained. It is nice story.

comments:

1, C1 position modification: the nitro-phenyl group did not have any contact with proten residures. But why its bonding affinity are higher than propargyl and methyl groups?

2, The O-3' modification: the sulfate group gave better bonding affinity. What is the reason? How about the phosphate or NH4 + group?

3, In page 4, scheme 2 (page line 120-122), could give the sideproducts for this glycosylation? The yield was 38% for the desired product.

4, In general procedures, please provide the details for final compounds' purification.

5, Format problems: there are many format problems in the manuscript, such as font-style, bold for compounds' numbers, space.

p1, line 2, 20....: N-acetyl should be N-acetyl, .....

p1 line 21....: 4-O- should be 4-O-;......

p3 line 106...: 60oC should be 60 oC; line 104 "8in" should "8 in"......

p4 scheme 2: HF.Py; HF-C5H5N

Et3N SO3 in scheme 2, Et3N.SO3 in scheme 3.

.........

Author Response

comments:

1, C1 position modification: the nitro-phenyl group did not have any contact with proten residures. But why its bonding affinity are higher than propargyl and methyl groups?

Reply:

As dicussed from the reviewer's 1 comments, solvation entropy contribution play a key role in protein-ligand interactions. In the present case, as seen with the -Tdelta-S term, the value of the delta-S term is more positive, hence more favorable. The aspect is included at the end of section 2.2

2, The O-3' modification: the sulfate group gave better bonding affinity. What is the reason? How about the phosphate or NH+ group?

Reply:

As clearly discussed in the crystallographic section (Fig. 2 and text), the combined interactions between the Arg144 and His158 are nicely positioned to form complementary and favorable anionic-cationic-H-bonding, which are further conterbalanced with bonded water molecules. 

3, In page 4, scheme 2 (page line 120-122), could give the sideproducts for this glycosylation? The yield was 38% for the desired product.

Reply:

This aspect was discussed in the reviwer 1 reply. We added comments to explain the low yield.

4, In general procedures, please provide the details for final compounds' purification.

Reply:

Good point, but the info was already included in (now) line 289-290. 

5, Format problems: there are many format problems in the manuscript, such as font-style, bold for compounds' numbers, space.

Reply:

We made all the detected corrections 

p1, line 2, 20....: N-acetyl should be N-acetyl, .....

p1 line 21....: 4-O- should be 4-O-;......

p3 line 106...: 60oC should be 60 oC; line 104 "8in" should "8 in"......

p4 scheme 2: HF.Py; HF-C5H5N

Et3N SO3 in scheme 2, Et3N.SO3 in scheme 3.